# Research Progress on Hypoglycemic Mechanisms of Resistant Starch: A Review

**DOI:** 10.3390/molecules27207111

**Published:** 2022-10-21

**Authors:** Jiameng Liu, Wei Lu, Yantian Liang, Lili Wang, Nuo Jin, Huining Zhao, Bei Fan, Fengzhong Wang

**Affiliations:** 1Key Laboratory of Agro-Products Quality and Safety Control in Storage and Transport Process, Ministry of Agriculture and Rural Affairs, Institute of Food Science and Technology, Chinese Academy of Agricultural Sciences, Beijing 100193, China; 2College of Food Science and Technology, Hebei Agricultural University, Baoding 071033, China

**Keywords:** resistant starch, diabetes, hypoglycemic mechanisms, intestinal microbiota

## Abstract

In recent years, the prevalence of diabetes is on the rise, globally. Resistant starch (RS) has been known as a kind of promising dietary fiber for the prevention or treatment of diabetes. Therefore, it has become a hot topic to explore the hypoglycemic mechanisms of RS. In this review, the mechanisms have been summarized, according to the relevant studies in the recent 15 years. In general, the blood glucose could be regulated by RS by regulating the intestinal microbiota disorder, resisting digestion, reducing inflammation, regulating the hypoglycemic related enzymes and some other mechanisms. Although the exact mechanisms of the beneficial effects of RS have not been fully verified, it is indicated that RS can be used as a daily dietary intervention to reduce the risk of diabetes in different ways. In addition, further research on hypoglycemic mechanisms of RS impacted by the RS categories, the different experimental animals and various dietary habits of human subjects, have also been discussed in this review.

## 1. Introduction

With changes in diet and lifestyle, the rising prevalence of diabetes has constituted one of the major threats to human health, globally. In the past thirty years, the number of people with diabetes has quadrupled, and diabetes has become the ninth leading cause of death [1]. It was estimated that the number of people with diabetes will reach 6.43 million worldwide by 2030. In addition, it cost at least USD 966 billion in global health expenditure in 2021, according to the International Diabetes Federation (IDF) [2], which showed an enormous global economic burden of diabetes. In the long term, lifestyle interventions, especially the changes in diet, have been suggested as the primary treatment for regulating the blood glucose level for patients. The consumption of easily digestible carbohydrates, such as sucrose and starch can affect the level of blood glucose directly, since they can be digested in the human gastrointestinal tract. RS is defined as a kind of starch that cannot be digested by amylases in the small intestine and eventually is fermented in the colon by microbiota [3]. There are relevant studies that show that consuming RS has a positive effect on regulating human blood glucose levels [4,5]. It has been confirmed that the glycemic response could be reduced by RS, compared with normal carbohydrates in an approved European Food Standards Agency claim (EFSA) [6].

As a special kind of dietary fiber, RS can’t be digested in the small intestine [7]. It can be divided into five categories (i.e., RS1, RS2, RS3, RS4, RS5), based on its different physical and chemical structures [8]. The relevant studies have reported the positive effects of RS in the regulation of type 2 diabetes mellitus (T2DM) [9,10]. There are also studies which found that the postprandial glycemic response could be controlled [11,12], the insulin sensitivity (IS) could be increased [13,14] and the expression of the inflammatory markers could be reduced by RS, as observed in clinical trials and in rodent models. However, the exact mechanism of how RS exerts its hypoglycemic effect has not been fully understood.

This review summarizes the current studies on RS in diabetes control and elaborates on the potential mechanisms (Figure 1, illustrating the structures of different types of, and the relevant hypoglycemic mechanisms of RS). At the same time, we summed up the limitations of the current research and provided a reference for future research perspectives.

## 2. Regulating the Intestinal Microbiota Disorder

The risk of development of T2DM could be prevented by RS through the gut microbiota, with the help of regulating the abundance of microbiota, to produce starch-degrading enzymes, and improving the intestinal barrier function [15]. Gut microbiota are composed of a variety of commensal microorganisms, including certain amounts of bacteria, fungi and viruses. They play an important role in regulating the metabolic, endocrine and immune functions [16].

Short chain fatty acids (SCFAs) are one of the important bio products of intestinal microbiota, mainly including acetic acid (C2), propionic acid (C3) and butyric acid (C4). These three SCFAs are also the most abundant SCFAs in the human body [16]. SCFAs are mainly produced from the fermentation of non-digestible carbohydrates (e.g., RS) [17]. They can improve the insulin resistance (IR) and T2DM, by regulating the related metabolic pathways. Different from the mechanisms that affect glucose homeostasis directly, SCFAs impact the host health at the cellular, tissue and organ levels [18].

SCFAs can promote the secretion of two important key intestinal hormones, namely the glucagon-like peptide-1 (GLP-1) and peptide YY (PYY). This secretion-boosting effect is able to increase the satiety by acting on the gut-brain axis. By this pathway, SCFAs can reduce appetite and food intake indirectly, which could prevent weight gain and thereby lower the risk of diabetes. SCFAs can also regulate the blood glucose concentrations by increasing the insulin secretion mediated by GLP-1 [19]. Nielsen et al. [20] found that, compared with the Western-style diet (WSD) group, there were 2- to-5-fold increases of butyrate pool size in the large intestinal digesta in the RS diet (RSD) and arabinoxylan diet (AXD) groups. They inferred that the result of stimulating the insulin secretion was caused by the promotion of the intestinal endocrinology of PYY, which inhibited the timing of the gastric and intestinal translocation. Then, the appetite was suppressed while the GLP-1 secretion was promoted. Hughes et al. [21] found that the fasting and peak concentration of peptide PYY3-36 increased while the peak concentration and AUC of the glucose-dependent insulinotropic peptide decreased after the healthy adult subjects ingested RS2-enriched wheat. Binou et al. [22] found that the bread rich in the β-glucans (βGB) groups and the bread rich in the RS (RSB) groups elicited a lower incremental area under the curve (AUC) for the glycemic response, compared with the control group (glucose solution, GS). At 15 min after the βGB and RSB intakes, a significant reduction in appetite and an increase in satiety were detected in the healthy adults, and this trend continued up to the 180th min. The result showed that the food containing RS could retard the absorption of glucose. Maziarz et al. [23] found that the total concentration of PYY in the high-amylose maize type 2 resistant starch (HAM-RS2) group was significantly higher than in the control group (*p* = 0.043) at 120 min. At the same time, the AUC glucose (*p* = 0.028) was decreased at the end of 6 weeks in the HAM-RS2 group, while this trend was not related to the changes in the subjects’ physical composition or total energy intake. This result might be caused by the SCFAs that are produced from the fermentation of HAM-RS2 by the bacteria in the lower GI tract. At the same time, the relevant studies have suggested that HAM-RS2 might show its benefits by increasing the SCFAs in the blood to alter the free fatty acid and glycerol that are released by adipocytes, regulate the bile acid metabolism [24,25] or alter the intestinal microbiota profile [26].

Mohr et al. [27] have reported similar results as they found that the postprandial blood glucose and insulin levels could be reduced by the combination of RS and whey protein. Thus, they inferred that RS could enhance the variety of metabolites in the gut, e.g., the production of the SCFAs could be improved. Furthermore, Zhou et al. [28] found that the plasma GLP-1 and PYY levels were both increased at different time points within a 24 h period in mice fed with RS (53.7% RS2, 10 d), and this result was not related to diets, the different glycemic indexes or the time of the blood sample collections. Chen et al. [29] found that the glucose of diabetic mice in three (corn, mung bean and Pueraria) RSs were significantly lower than the diabetic groups, after 14 weeks. The GLP-1 content of diabetic mice, at 19 weeks, showed significant differences in corn RS groups and mung beans RS groups (*p* < 0.01). Chen et al. [30] found that the serum blood glucose level of mice in the high-dose multiple composite RS group was reduced by 59.71%, compared with the model control group, indicating that the blood glucose could be controlled by multiple compounds effectively, and the effect of the high-dose multiple composite RS on reducing the blood glucose was better. Boll et al. [31] found that arabinoxylan oligosaccharides (one of the dietary fibers, AXOS) showed the ability to improving the glucose tolerance in an overnight perspective. The possible mechanism was that IS and the gut fermentation could be improved by the breads containing an AXOS-rich wheat bran extract and RS, separately or combined, on the glucose tolerance and the intestinal markers in healthy subjects.

Many studies found that the concentration of SCFAs C2, C3, C4 could be promoted by RS, leading to the decrease of the pH in the intestine. The falling pH would promote the production of the beneficial bacteria and reduce the number of intestinal spoilage bacteria, achieving a balanced state of the intestinal microbiota [32,33]. Zhang et al. [34] found that the blood glucose could be reduced, the response to the IR and the glucose tolerance test could be ameliorated, and the pathological damage could all be relieved by RS3, in T2DM mice, from the *canna edulis* (Ce-RS3). In this study, 24 diabetic mice, induced by streptozotocin (STZ,) were randomly divided into a T2DM group (Model), a RS group (Ce-RS3) and a metformin group (Met). Eleven weeks later, they found the microbial and metabolic disorder of the mice in the RS and Met groups were significantly regulated. Among them, Ce-RS3 showed a better regulatory effect and an improved diversity of the intestinal microbiota, especially of the *Prevotella* genera. The SCFAs levels were significantly increased, since the abundance of the gut bacterial producing SCFAs was increased, such as *Phascolarctobacterium*, *Ruminococcaceae_NK4A214_group*, *Ruminococcaceae_UCG_014*, *Helicobacter* and *Ruminooccus*. Therefore, they inferred that the intestinal microbiota characteristics of the RS group were closely associated with the T2DM-related indicators. Zhou et al. [35] found that the intestinal flora microbiota abundance was regulated by the intake of BRS (Buckwheat-RS), which increased the abundance of the beneficial bacteria *Lactobacillus*, *Bifidobacterium* and *Enterococcus*, while the abundance of *Escherichia coli* decreased. Compared with the HFD (high-fat diet) group, the content of the SCFAs in the mice colons was increased in the BRS group. In this study, the male C57BL/6 mice were fed a normal diet (CON), HFD, and HFD supplemented with BRS (HFD + BRS) for 6 weeks, separately. The quantities of four common and major intestinal microbiota (*Bifidobacterium*, *Lactobacillus*, *Enterococcus*, *E. coli*) were analyzed by qPCR and the absolute quantification methods. It has been speculated that the changes in the intestinal microbiota caused by the BRS, might be related to its ability to regulate the intestinal redox status. Sánchez-Tapia et al. [36] found that the RS in black beans could improve the glucose response, because the gut microbiota, such as *C**lostridia,* could be mediated by the black beans’ RS. Zhu et al. [37] found that the gut microbiota composition in the T2DM mice, changed obviously. The abundance of the genus *Clostridium* and *Butyricoccus* could be increased, while the genus *Bacteroides*, *Lactobacillus*, *Oscillospira* and *Ruminococcus* could be decreased by the ORS (oat RS). In addition, the Pearson correlation analysis showed that the genus *Bacteroides*, *Butyricoccus*, *Parabacteroides*, *Lactobacillus*, *Oscillospira*, *Ruminococcus* and *Bifidobacterium* were positively correlated with the occurrence of diabetes and inflammation (*p* < 0.05), while genus *Clostridium* and *Faecalibacterium* showed a negative correlation (*p* < 0.05). The result indicated that the anti-diabetic effects of the ORS was achieved by altering the gut microbiota.

Additionally, the metabolites of intestinal microbiota can improve the intestinal barrier, reduce the IR and the expression of the related inflammatory factors [38]. Jiang et al. [39] found that, compared with the NC group (normal control, normal mice on a basal diet) and the MC group (model control, diabetic mice on a basal diet), *Firmicutes* and *Bacteroidetes* were the dominant bacterial phyla in the IG group (intervention group, the diabetic mice fed with of Ganoderma lucidum spores encapsulated within the RS (EGLS)). The abundance of *Proteobacteria* (mostly identified as pathogenic bacteria) in diabetic mice was the highest. The elevated level of *Proteobacteria* might indicate the intestinal inflammation in the MC group, which might be related to the occurrence of T2DM. However, compared with the MC group, the proportion of *Proteobacteria* in the IG group was significantly reduced. Therefore, they speculated that the blood glucose in mice was decreased since the fecal microbial community abundance associated with promoting the anti-inflammatory responses were modulated by EGLS. Kingbeil et al. [40] found that, compared with the low-fat chow (LF, 13% fat) and the HF (45% fat) intervention, the isocaloric HF supplemented with a 12% potato RS (HFRS) intervention in the HF-fed mice, would lead to changes in the composition of the gut microbes. They found this result correlated with the improved inflammatory status and the vagal signaling by the potato RS. Beyond that, they found that the energy consumed by the HFRS-fed mice was significantly less, compared with the HF-fed mice. Additionally, the systemic inflammation and the glucose homeostasis were improved in the HFRS group, compared to the HF group. Another study [41] showed that the improved intestinal barrier function in the potato RS treated mice was associated with the reduced systemic inflammation and the improved glucose homeostasis. For the HFD mice, the intestinal barrier function was decreased and the inflammation responses were initiated because of the gut microbiota dysbiosis. However, the RS supplementation could increase the SCFAs production that might decrease the effects of the HFD by enhancing the gut barrier function, reducing the levels of the systemic lipopolysaccharide (LPS) and increasing GLP-1 levels. In addition, the IR could be sufficiently promoted by the chronic elevation of the circulating LPS.

Keenan et al. [8] found that, in human subjects, the IS was increased after consuming the RS. However, only one of several studies reported an increase in the serum GLP-1 associated with RS added to the diet. This means that RS might reduce the blood glucose through other mechanisms, such as the increased intestinal gluconeogenesis, which might be associated with the promotion of the improved IR. Indeed, there were several studies suggesting that the SCFAs could decrease the hepatic glycolysis and gluconeogenesis but increase the glycogen synthesis [42,43,44].

According to the above studies, it can be found that the mechanisms of RS, based on the intestinal hypoglycemia, could be divided into three categories. Firstly, the RS could be fermented into intestinal metabolites related to the regulation of the blood glucose, such as the SCFAs. Secondly, the abundance of the beneficial bacteria, such as *Lactobacillus*, *Bifidobacterium*, *Enterococcus* and *Ruminococcus* could increase, while the abundance of the harmful bacteria, such as *Bacteroides*, *Lactobacillus*, *Oscillospira* and *Escherichia coli* would decrease by the RS to regulate the metabolic pathway of the blood glucose. Thirdly, RS could decrease the expression of the inflammation factors, such as tumor necrosis factor-α (TNF-α), and interleukin-6 (IL-6). Additionally, the hepatic glycolysis and gluconeogenesis could be decreased by the SCFAs fermented by the gut microbes. Table 1 illustrates the reported hypoglycemic mechanisms of RS by regulating the intestinal microbiota disorder.

## 3. Resisting Digestion

It has been shown that RS could regulate the levels of glucose and insulin in vivo and be beneficial to maintain the homeostasis of glucose. Due to its metabolic characteristics of slow absorption, RS plays a significant role in controlling and intervening in the condition of diabetes by reducing fasting and the postprandial blood glucose, as well as increasing the IS [45].

Bindels et al. [46] have shown that the increase of the insulin level mediated by RS also occurred in the absence of the relevant microbiota, through parallel experiments on RS fed conventional mice and sterile mice. The cecal concentrations of several bile acids (BAs) were changed, and the gene expression of the macrophage markers was reduced in the adipose tissue, of which the polarization phenotypes was implicated in the control of IS in both mice groups. The result showed that both the IS and the glucose homeostasis could be regulated by the BAs via the nuclear farnesoid X receptor (FXR) and the membrane-bound TGR5 signaling.

Wang et al. [47] found that the average blood glucose and the postprandial blood glucose could be reduced significantly in T2DM patients, with the blood glucose fluctuations decreasing after the RS diet treatment and the oral administration of glucose. The results were preliminarily inferred to be related to the anti-digestion characteristics of RS. Strozyk et al. [48] found that, compared with the fresh rice (NR) group, the peak of the postprandial blood glucose in type 1 diabetes was lower in the cooling and reheated rice (CR) group. A shorter time of the glycemic peak has also been observed in the CR group, suggesting a beneficial effect to the glycemic control, as the delayed glycemic peak could improve the activity of the short-acting insulin analogues. Yadav et al. [49]] have also found that the content of RS was increased in starch products with multiple heating/cooling cycles, while the content of digestible carbohydrates was reduced. Haini et al. [50] found that, compared with the control group, the 2-h postprandial glucose of healthy female subjects was lower in the high-amylose maize starch 30 (HM30) group. In the HM group, 30% wheat flour has been replaced by HM in a Chinese steamed bun (CSB), which decreased the content of the digestible starch and the digestion speed of the starch. Therefore, the glycemic response and the increase in the postprandial blood glucose of healthy adult subjects have been delayed. Djurle et al. [51] found similar results, a slower rise of the postprandial glucose in healthy adult subjects was observed in the RS bread group. In this group, the breads were made with refined flour containing RS. Maki et al. [52]. have assessed the effects of the two doses of HAM-RS2 intake on the IS participants with different waist circumferences. The participants were randomized to receive 0 (control starch), 15, or 30 g/d (double-blind) of HAM-RS2 for four weeks with washout intervals of three weeks. At the end of each period, the minimal model IS had been evaluated by using an insulin-modified intravenous glucose tolerance test. The present results suggested that the intake of HAM-RS2 at 15–30 g/d could improve IS in obese men whereas no significant change in IS was observed in women for reasons that remain to be determined. Zeng et al. [53]. found that the type 3 resistant starch (RS3) couldn’t be degraded into glucose by the digestive enzymes in the human intestine, which could reduce the amount of the glucose conversion by the human body. The RS3 could also reduce the glycemic index that helped to reduce the postprandial blood glucose. At the same time, Wang’s study [54] has shown that RS3 could stabilize the human blood glucose by repairing the pancreas β cell function, as well as improving the IS and IR of the peripheral tissues. Gourineni et al. [55] have completed a study on type 4 resistant starch (RS4). In this study, a nutritional bar containing a control (2 g), medium (21 g) and high (30 g) fiber, were consumed by healthy adults (*n* = 38). Venous glucose, insulin, and the capillary glucose were measured at the end. They found that the concentrations of the capillary glucose and venous insulin in the two fiber groups were significantly lower than those in the control group. At the same time, they found that the postprandial glucose and insulin responses were significantly reduced in the generally healthy adults who consumed the bar containing the potato RS4 fiber.

There are also several other studies about RS4. Stewart et al. [56] have proved that substituting RS4 for a digestible carbohydrate in scones significantly lowered the blood glucose levels in healthy adults. Likewise, Mah et al. [57] have replaced the digestible starch with cassava RS4, to reduce the available carbohydrates and they found that the postprandial blood glucose and insulin concentrations decreased significantly in the healthy subjects. Other studies also found similar results by using RS4 (25 g of VERSAFIBE™ 1490 (Ingredion Incorporated, Bridgewater, NJ, USA)) to replace the normal starch in cookies [58]. In general, there was a study that showed that glycemia could be reduced by replacing the rapidly digestible starch with RS4. This result might be caused by the incomplete release of glucose and the anti-digestibility of the starch [59]. Wang et al. [60] have postulated that the diabetes-related liver glycogen fragility could also be attenuated by RS. They found that both the diabetic group and the non-diabetic group of mice, fed with two types of high-amylose RS, contained less hepatic glycogen than those fed with normal corn starch (NCS). In addition, the molecular size and the chain-length distributions of the liver glycogen were characterized to detect the fragility of the liver glycogen before and after the dimethyl sulfoxide (DMSO) treatment. The result showed that the high-amylose RS diet could prevent the fragility of the liver-glycogen α particles, which were consistent with the hypothesis that hyperglycemia was related to the glycogen fragility. They postulated the reason was that the high-amylose RS was eventually fermented in the large intestine rather than in the small intestine, which elicited beneficial effects on the glycemic response and T2DM.

Through the above studies, it is not hard to find that the IS could be affected by RS through the reducing gene expression of the macrophage markers in the adipose tissue, regulating the membrane-bound TGR5 signaling, repairing the pancreas β cell function and preventing the fragility of the liver-glycogen α particles. Meanwhile, the RS shows less effect on the blood glucose, since it cannot be degraded by the digestive enzymes in the small intestine but only be fermented in the large intestine that reduces the absorption of glucose.

## 4. Reducing Inflammation

Studies have shown that the damaged pancreatic β cells could be repaired, while the expression of the binding genes, such as the C-reactive protein (CRP), TNF-α and interleukin, could be down-regulated by the RS to show the hypoglycemic effect [61].

Gargari et al. [62] found that the glycated hemoglobin (HbA1c) (−0.3%, −3.2%) and TNF-α (−3.4 pg/mL, −18.8%) could be decreased by the RS2, compared with the placebo groups. In this study, 28 females with diabetes took RS (intervention group) and 32 took a placebo (placebo group) at 10 g/d for 8 weeks. The fasting blood sugar (FBS), HbA1c, lipid profile, high-sensitive CRP (HS-CRP), IL-6 and TNF-α were measured at the end of the trial. The results suggested that the glycemic status and the inflammatory markers in the women with T2DM could be improved. Based on the results, they speculated that the improvement in the glycemic status was due to the reduction of the TNF-α levels. And Tayebi Khosroshahi et al. [63] came to similar conclusions through their research. They found that the IR level and the body’s IS could be improved by RS. In the study, a 20–25 g high linear chain RS and wheat flour, daily, were used to treat hemodialysis patients for 8 weeks, respectively. The results showed that the serum IL-6 and TNF-α levels in the RS group were significantly decreased.

Xu et al. [64] found that the blood glucose of obese mice could be reduced efficiently by RS. The obese mice were placed into four groups: NC, HF, URS (intervention group with RS from untreated lentil starch) and ARS (intervention group with RS from autoclaved lentil starch). The mice in the ARS and URS groups were administrated intragastrically with the ARS and URS (400 mg/kg·BW) suspension, once daily. Furthermore, the histological analysis and the gut microbiota analysis suggested the results above might be achieved, based on the improvement of the inflammatory state and the changes of the microbial components related to vagal signals. Yuan et al. [65] have reported the similar results. Compared with the normal rice (NR)-treated diabetes mice, the levels of the related inflammation factor, such as the serum CRP, TNF-α, IL-6, nuclear factor-k-gene binding (NF-κB) and leptin (LEP), were lower while the Adiponutrin (ADPN) level was higher in the selenium-enriched rice with a high RS content (SRRS) treated mice and the normal rice with the high RS content (NRRS) treated mice. The results suggested that the hypoglycemic effects might be achieved by the high RS rice treatment because of the improvement of the chronic inflammation.

It is not hard to see that the levels of the related inflammatory factors, such as CRP, IL-6, TNF-α and NF-κB, were lowered by the RS. The reduction of glomerular damage and the enhancement of the glomerular reabsorption alleviated the development of diabetes.

## 5. Regulating Hypoglycemic Related Enzymes

The level of the blood glucose could be regulated by some metabolic enzymes, such as glycogen synthase (GS), phosphoenolpyruvate carboxy kinase (PEPCK) and α-glucosidase. The activity of these enzymes could be regulated by the RS to achieve a hypoglycemic effect.

Zhou et al. [66] found that the blood glucose level in the RS administration group diabetic mice was lower than that in the control group. Moreover, the expression of the insulin-induced genes Insig-1 and Insig-2, that were related to the glycolipid metabolism, were also significantly up-regulated after the RS administration in mice. The blood glucose level in the diabetic mice fed with RS was regulated by promoting glycogen synthesis and the inhibiting gluconeogenesis. Further studies suggested that the expression level of isoform 1 of the glucose-6-phosphatase (G6PC1) catalytic subunit, was lower in the RS group than it was in the MC group. In addition, this study found that the expression of the glycogen synthesis genes, the GS and glycogenin1 (GYG1) increased more than twofold after the RS intake, which suggested a progressive stimulation of the hepatic glycogen synthesis in the liver. These results suggested that the inhibition of the gluconeogenesis and the promotion of the glycogen synthesis may be one of the main ways for RS to decrease the blood glucose. This study demonstrated that the mRNA encoding enzymes involved in the gluconeogenesis could be reduced by the RS to alleviate the glucose metabolic disorders in diabetic mice. Zhu et al. [67] found that after the intervention with a kind of RS in banana powder, the glucose uptake in the liver, the glycogen synthesis, the IS and IR of the db/db diabetes mice, were improved, while the mRNA expression of the key enzyme PEPCK, the carbohydrate response element binding protein (ChREBP) of the gluconeogenesis and the GSK-3 of the glycogen synthesis, were all significantly down regulated by the RS. Hao et al. [68] found that the green banana powder was rich in RS2 and made biscuits from it, which verified its feasibility. Xiao et al. [69] found that the blood glucose of T2DM Kunming (KM) mice was increased by 10.9% in the control group, while it was decreased by 14.7% in the RS group. The inhibition rate of α-glucosidase that related to the blood glucose peak in the T2DM mice, was measured. In RS group, the inhibition rate was 23.13%, showing a certain inhibitory effect. Since the activity of α-glucosidase could be inhibited by the RS, the consumption of the liver glycogen would be reduced and the trend of weight loss would be alleviated as well.

Above of all, it’s not difficult to find that the present studies of the related enzymes were all carried out in mice. In addition, to reduce the blood glucose, the expression level of the key enzymes, such as GS, G6PC1, PEPCK, ChREBP, GK and α-glucosidase, could be lowered by the RS treatment, leading to the reduction of IS and IR.

## 6. Other Mechanisms of the Hypoglycemic Action

In addition to the above mechanisms, there are some other pathways of RS that play a role in hypoglycemia.

Li et al. [70] found that the blood glucose of the model group was significantly increased, compared with the control group (*p* < 0.05). In this study, the hyperglycemia mice induced by HFD, were treated with a dioscorea alata L. high RS (HRS) for 4 weeks. The blood glucose increased since the ability of converting the glucose into lipids was weakened because of the disorder of the fat metabolism. Li et al. [71] found that the blood glucose of the mice with diabetes was regulated after the intake of the biscuit with the added RS3 from *Purple Disocorea Alata*. L. Wang et al. [72] found the colonic proglucagon expression and the adiponectin levels in visceral fat could be increased by HAMRS2, which indicated that the IS in the visceral fat has been improved. Sun et al. [73] found that the glucose tolerance, the insulin content and glucose metabolism in diabetic mice were regulated by the RS2 treatment. In the study, they treated the T2DM mice with a high-glucose-fat diet and a low-dose STZ with low, medium, and high doses of RS2 (100, 150, and 200 g/kg) for 28 days. Furthermore, the western blot and real-time polymerase chain reaction (RT-PCR) results showed that the expression levels of the insulin receptor substrate 1 and the insulin receptor substrate 2, were enhanced in the pancreas. Based on the above results, the blood glucose in the diabetic mice can be regulated by the RS by altering the expression level of the genes related to the glucose metabolism and improving the pancreatic dysfunction. Wang et al. [74] found that the blood glucose level was reduced by 16.0–33.6% and the serum insulin level was recovered by 25.0–39.0% in T2DM mice fed on a lotus seed RS (LSRS). They elucidated the molecular basis of the hypoglycemic effect by supplying different doses of the LSRS on the T2DM mice. Through the relevant analysis of genes, they have suggested that the protective effect of the LSRS was most likely achieved by modulating the expression levels of the various key factors involved in the insulin secretion, insulin signal transmission, cell apoptosis, antioxidant activity and p53 signaling pathways.

MacNeil et al. [75] found that, as an effective substitute for the available carbohydrate (CHO) in baked food, RS could lower the T2DM diabetes’ blood glucose excursion by using a randomized crossover design. Furthermore, the GIP-insulin axis was influenced after ingesting more RS because of the hyperglycemic effect of the RS. In this study, 12 patients with T2DM underwent four different bagel treatments. Abby et al. [76] have found that the glucose, insulin and glucagon-like peptide-1 have been reduced significantly in the fasted subjects. In the study, the fasted subjects (*n* = 20) consumed either a low-fiber control breakfast or one of four breakfasts that contained a 25 g soluble corn fiber (SCF) or RS, alone or in combination with pullulan (SCF + P and RS + P). The results suggested that the satiety or the energy intake would not be influenced by the fiber treatments, compared to the control. Though the definite mechanism of the results haven’t been described, it may be related to the secretion of GLP-1, and the aging-related decline in the glucose tolerance could be recuperated by it [77]. Song et al. [78] found the value of the fasting blood glucose of T2DM mice was decreased by investigating the effects of the Kudzu RS on the IR, the gut physical barrier and the gut microbiota. The expression of IRS-1, p-PI3K, p-Akt, and Glut4 were restored by the study of the relevant expression of the protein, which led to the improvements of the insulin synthesis efficiency and the glucose sensitivity in the T2DM mice.

In conclusion, other mechanisms of the RS hypoglycemic actions could be divided into three categories. Firstly, the disorder of the fat metabolism could be decreased by the RS. The tryptophan related to the gut microbiota function and the IS in the visceral fat could be regulated by the HRS. Secondly, the expression levels of the genes related to glucose, such as the insulin receptor substrate 1 and the insulin receptor substrate 2, could be enhanced in the pancreas by the RS. The expression levels of the various key factors involved in the insulin secretion, insulin signal transmission, cell apoptosis, antioxidant activity and p53 signaling pathways could also be modulated by the RS. Thirdly, the expression of protein, such as IRS-1, p-PI3K, p-Akt, and Glut4 could be restored by the RS. All in all, the relevant studies still have some limitations and need further study.

## 7. Conclusions

According to numerous previous studies, RS has been confirmed as a kind of dietary fiber to prevent diabetes. In this review, several mechanisms of the glycemic control with a RS consumption were summarized, mainly including regulating the intestinal microbiota disorder, resisting digestion, reducing the inflammation and regulating the hypoglycemic related enzymes. Several specific intestinal microbiota, signaling pathways, gene targets and relevant enzymes of those mechanisms have been clarified in the above studies. Therefore, RS seems to hold great promise in the prevention and treatment of diabetes. Based on the above research, we have concluded the studies on the prevention of T2DM by RS (as shown in Table 2).

However, previous studies are incomplete, since most of the studies have been focused on animal experiments, rather than on human subjects. To our knowledge, the relevant literatures on animal experiments are mainly focused on mice, but a few studies are on large animals. Although there are a couple of relevant studies on pigs, the change of the PYY levels showed different results, compared with the mice. It is indicating that the role of PYY on the RS in different animals is controversial. Thus, further investigations are needed.

In addition, even the reports on the human subjects are not comprehensive. There are several factors leading to inaccurate conclusions. Firstly, the RS has been classified into five main categories, according to the causes of indigestion. They process the different molecular structures and amylase binding sites. Meanwhile, the RS molecules are linked together by different glycosidic bonds that cause diverse effects on the blood glucose. Therefore, it is necessary to clarify the mechanisms about how the blood glucose is influenced by the different RS structures for further studies, especially in the changes of the different enzymes related to the different RS structures. Secondly, the current conclusions obtained in mice are not necessarily applicable to human subjects, since the metabolic pathways and targets are different. Therefore, more thorough studies on humans should be conducted. In order to clarify the specific metabolites or the proteins that are related to the blood glucose in the human body, such studies should not only observe the level of blood glucose, but also analyze the metabolites in the blood and urine by metaomics, such as metatranscriptomics and metaproteomics. With the help of the statistical analysis, the main pathways, key enzymes and target genes related to the hypoglycemic activity in humans will be screened out. Thirdly, the various dietary habits may also cause the relevant genetic changes in the same ethnic groups, which may lead to the different responses to the same RS. There are certain differences in the intestinal microbiota and metabolic pathways among people with different dietary habits. For example, the abundance of some harmful intestinal microbiota, such as *Bacteroides* and *Lactobacillus*, will be increased in people with a high animal protein or high animal fat diet, while the abundance of several intestinal microbiota that produce SCFAs, such as *Helicobacter*, *Akermanniella* and *Bifidobacterium* will be increased in people with a high dietary fiber diet. In addition, there are also studies that suggest that the postprandial cardiovascular and metabolic indexes are different in the equal-energy diets with different proportions of macronutrients, such as fiber, fat and protein, as well as influenced by body weight and exercise. To date, data from the intervention studies that systematically assess the effects of the different diet habits on glucose and metabolomics, are particularly lacking. Therefore, to avoid the interference of the results on the RS hypoglycemic activity, a long-term analysis of the dietary habits, should be carried out in future studies.

In conclusion, the mechanisms of the RS hypoglycemic activity in the human body need to be further studied from the aspects discussed above. Those studies can help us to understand the mechanisms comprehensively, as well as to provide the theoretical basis for people to choose a specific type of RS to control their blood glucose.

## Figures and Tables

**Figure 1 molecules-27-07111-f001:**
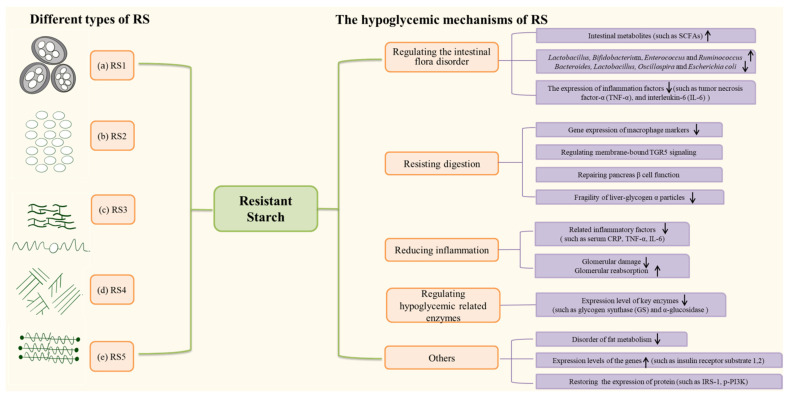
Different types of, and the potential hypoglycemic mechanisms of RS. (**a**) RS1, the most common RS found in all grains, is a kind of physically inaccessible starch.; (**b**) RS2, in raw potato or high-amylose maize, has a B- or C-type polymorph; (**c**) RS3, in cooked and cooled potatoes, a kind of retrograded starch; (**d**) RS4, chemically modified starch, through the addition of cross-linkages or chemical derivatives; (**e**) RS5, in processing or cooking oil, a lipid-modified starch. Adapted from Wong et al. [10].

**Table 1 molecules-27-07111-t001:** The reported mechanisms of regulating the intestinal microbiota disorder.

Type of RS and Its Source	Model	Dosage/Duration	Intestinal Hormone/Intestinal Microbiota	Inferences	Ref.
76% HAM-RS2 (high-amylose maize) +24% raw potato starch	Thirty female pigs (BW 63.1 ± 4.4 kg)	RSD and AXD diets: 2.7% of average BW (75 kg); WSD diet: 2.44% of BW/3 weeks	Increase PYY	PYY promoted intestinal secretion, promotes GLP-1 secretion and stimulates insulin secretion	[19]
Bread enriched with resistant starch (RSB) (15% of total starch)	Ten apparently healthy subjects (mean 27 years; SD 3.9) with a normal body mass index (mean 24.5 kg m^−2^; SD 2.8)	An amount corresponding to 50 g of available carbohydrates or a solution containing 50 g of glucose diluted in 250 mL of water/Test sessions, total 4 weeks.	Increase GLP-1 and PYY	The food contains RS and could retard the absorption of glucose	[20]
HAM-RS2(high-amylose maize type 2 resistant starch)	Eighteen overweight, healthy adults	Either muffins enriched with 30 g HAM-RS2 (*n* = 11) or 0 g HAM-RS2 (control; *n* = 7) daily/6 weeks	Increase PYY	The consumption of HAM-RS2 can improve glucose homeostasis, lower leptin concentrations, and increase fasting PYY	[21]
Pancake with RS	Eight healthy, adult man, middle-aged (51.4 ± 11.5 years), normal- and over-weight (BMI = 29.84 ± 7.77 kg/m^2^; percent body fat = 26.42 ± 11.62%)	Consumed together with water (180 mL) within 12 min	Increase SCFA production	Combination of the RS and WP might enhance the gut SCFA production and reduce the blood glucose	[25]
RS (Hi-Maize 260)	One hundred adult male Sprague–Dawley rats	On the basis of the amount of Hi-Maize (56% RS) used/10 days	Increase GLP-1 and PYY	The plasma GLP-1 and PYY levels that regulate blood glucose were increased	[26]
Corn, mung bean and Pueraria RS	Fifteen diabetic rats induced with STZ	19 weeks	Increase GLP-1	The GLP-1 show a different content, the level of it might be related to the level the blood glucose	[27]
Ce-RS3 (RS3 from canna edulis)	Twenty-four diabetic mice induced with STZ	2 g/kg/11 weeks	Improve *Phascolarctobacterium*, *Ruminococcaceae_NK4A214_group*, *Ruminococcaceae_UCG_014, Helicobacter* and *Ruminooccu*; Decrease *Streptococcus* and *Bacillus genus*	The gut microbial properties of the RS group were tightly associated with the T2DM-related indexes	[32]
BRS (Buckwheat-RS)	Twenty-seven male 4-week-old C57BL/6 mice	6 weeks	Increase *Lactobacillus*, *Bifidobacterium* and *Enterococcus*; Decrease *Escherichia coli*	The gut microbiota change caused by BRS might be associated with the capacity of regulating the gut redox status	[33]
ORS (oat RS)	Fifty male Sprague–Dawley rats (4 weeks old, WD 105 ± 10 g)	6 weeks	Increase *Clostridium* and *Butyricoccus*; Decrease *Bacteroides*, *Lactobacillus*, *Oscillospira* and *Ruminococcus*	The anti-diabetic effects of the ORS were achieved by altering the gut microbiota	[35]
RS (Hi-maize TM)	Twenty-four healthy Sprague–Dawley rats (male, 190 ± 10 g weight)	10.5 g/kg bw/day/28 days	Increase *Proteobacteria*	The reduction in the blood glucose might be related to the changes in the fecal microbial community which promoted an anti-inflammatory response	[37]

**Table 2 molecules-27-07111-t002:** Studies on the prevention of T2DM by RS.

Kind of RS	Results	Conclusion	Ref.
RS2	The glycemic status and inflammatory markers in women with T2DM could be improved.	The improvement in glycemic status was due to the reduction of the TNF-α levels.	[62]
RS2	The expression levels of the insulin receptor substrate 1 and the insulin receptor substrate 2 were enhanced in the T2DM mice.	RS could regulate the expression level of the genes related to the glucose metabolism and improving the pancreatic dysfunction.	[73]
Lotus seed RS (LSRS)	The blood glucose level was reduced by 16.0–33.6% and the serum insulin level was recovered by 25.0–39.0% in the T2DM mice.	The LSRS achieved the hypoglycemic effect by modulating the expression levels of the various key factors.	[74]
Kudzu RS	The value of the fasting blood glucose of the T2DM mice was decreased.	Kudzu RS restored the expression of the relevant protein and it led to the improvements of the insulin synthesis efficiency and the glucose sensitivity in the T2DM mice.	[78]
Bagel with high-amylose maize RS (RS2)	The fasting IS of the RS bagel treatment is lower than the control bagel treatment.	The amount of insulin required to manage the postprandial glucose were reduced by the high-HAM-RS2 bagel through the improvement of the glycemic efficiency, while improving the fasting IS in adults at an increased risk of T2DM.	[79]
Indica rice resistant starch (IR-RS) prepared by modification	The blood glucose of the rats with T2DM was lower than those in the control group.	The IR-RS digestibility was affected as well as the blood glucose levels of the diabetic mice	[80]
White wheat flour bread (WWB) enriched RS	The glucose tolerance and GLP-1 were improved, compared with that without WWB.	The consumption of RS might affect the glycemic excursions through a mechanism involving colonic fermentation.	[81]
Banana starch (NBS) with a high resistant starch (RS)	The 24 h mean blood glucose (24 h MBG) value of the T2DM patients was lower in the NBS treatment but not significant.	The result might be influenced by different baseline microbiota, an underlying dietary variability, or other environmental factors.	[82]

## Data Availability

Not applicable.

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
