# Peer review of "Research Progress on Hypoglycemic Mechanisms of Resistant Starch: A Review"

_molecules, 2022, doi:10.3390/molecules27207111_

Round 1

Reviewer 1 Report

The article is interesting however the authors should include more references 2022, because in the database there are interesting articles in this area of research.

The authors should add a table with the studies on development of hypoglycemic foods and the prevention with resistant starch of type II diabetes.

I recommend that authors should include a section on future trends.

Reviewer 2 Report

Dear Authors,

The manuscript titled “Research Progress on Hypoglycemic Mechanisms of Resistant Starch: A Review” aims to summarize the important role of resistant starch in diabetes control, studying the different mechanism implicated.

The topic is very interesting, but English language must be extensively improved.

I recommend major revision.

Here are my remarks that hopefully would help to improve the article:

I suggest change the term “flora” by “microbiota” along the manuscript.

I suggest use the expression “reduce the risk of diabetes” instead of the term “prevent diabetes”.

Line 23. Please, revise if “hyperglycemic mechanisms” is a correct keyword, or whether, on the contrary, authors would want to say “hypoglycemic mechanisms”.

Line 31. Please, place a comma (“,”) between “2021” and “according”, and between “International Diabetes Federation (IDF) [2]” and “which”.

Line 32. Please, change “showing” by “showed”.

Line 36-37. For a better understanding, I recommend rewrite the following sentence: “While RS can resist being digested in stomach and small intestine because of its special structure”

Line 38-39. Please, rewrite the sentence (There are relevant studies, which showed that the intake of RS has a positive effect on regulating human blood levels…). Human blood levels of what? Please, specify it.

Line 40-41. It is necessary indicate that the stated health claim has been approved by Commission Regulation (EU) No 432/2012. It would be interesting to include the conditions of use of the claim “Replacing digestible starches with resistant starch in a meal contributes to a reduction in the blood glucose rise after that meal”.

Line 45-46. Please, insert “which” between “studies” and “found” (There are also studies which found…).

Line 50-53. Please, for a better understanding, I suggest authors rewrite the objective of their work.

Figure 1 must be cited in the text. Moreover, I recommend improve the format, as the text included in purple boxes is not legible.

Line 61-63. Please, rewrite it for a better understanding (The risk of development of T2DM could be reduced by RS…)

Line 78. According to the reference list, the cite number 19 is not Tina et al. Please, revise it.

Line 84. According to the reference list, the cite number 20 is not Panagiota et al. Please, revise it.

Line 84-86. Please, explain better the referred study.

Line 90. According to the reference list, the cite number 21 is not Mindy et al. Please, revise it.

Line 94-95. Please, include “are” between “that” and “produced” (…SCFAs that are produced from…).

Line 99. According to the reference list, the cite number 25 is not Alex et al. Please, revise it.

Line 99. Please, include “as” between “results” and “they”.

Line 99-102. Please, explain better the referred study.

Line 101. Do authors want to say, “enhance gut microbiota”?

Line 105-107. Please, explain better the referred study (Chen et al.)

Line 109. According to the reference list, the cite number 28 is not Liu et al. Please, revise it.

Line 109-111. Please, explain better the referred study.

Line 114. According to the reference list, the cite number 29 is not Elint et al. Please, revise it.

Line 122. According to the reference list, the cite number 32 is not Chi et al. Please, revise it.

Line 122-134. Please, explain better the referred study.

Line 145. According to the reference list, the cite number 34 is Sánchez-Tapia et al.

Line 146. Costridia is OK? Please, revise it.

Line 149. The genus of bacteria must be written by using italics letters. Please, revise it along the manuscript.

Line 165-168. For a better understanding, please, rewrite this sentence.

Line 168. According to the reference list, the cite number 38 is not Elizabeth et al. Please, revise it.

Line 191-195. For a better understanding, please, rewrite this paragraph.

Table 1 must be cited in the text. The name of the column “intestinal bioproduct/ intestinal microbio” is ok?. Please, if it is possible, include the dosage used in all studies included in the Table.

Line 218. Postpandrial blood glucose?

Line 221-222. According to the reference list, the cite number 46 is not Kevin et al. Please, revise it.

Line 228. Please, revise the author’s name (Zen or Zeng?)

Line 234. According to the reference list, the cite number 49 is not Vishnupriya et al. Please, revise it.

Line 237. Please, revise if the term “venous glucose” is correct.

Line 237-241. For a better understanding, please, rewrite this paragraph.

Line 248-249. For a better understanding, please, rewrite this sentence.

Line 272. According to the reference list, the cite number 56 is not Baharam et al. Please, revise it.

Line 279. According to the reference list, the cite number 57 is not Hamid et al. Please, revise it.

Line 355. According to the reference list, the cite number 66 is not Qi et al. Please, revise it.

Line 363. According to the reference list, the cite number 67 is not Stacey et al. Please, revise it.

Round 2

Reviewer 2 Report

Dear Authors,

The manuscript titled “Research Progress on Hypoglycemic Mechanisms of Resistant Starch: A Review” aims to summarize the important role of resistant starch in diabetes control, studying the different mechanism implicated.

The topic is very interesting, and authors have improved the manuscript.

I recommend minor revision.

Here are my remarks that hopefully would help to improve the article:

Line 36. Please, insert a space between “RS” and “is”

Line 87. According to the reference list, the cite number 21 is not Riley et al. Please, check it.

Line 289. Please, check the author’s name of reference number 62. Gargar or Gargari?
